# In situ single-shot diffractive fluence mapping for X-ray free-electron laser pulses

Michael Schneider[1], Christian M. Günther [1,2], Bastian Pfau [1], Flavio Capotondi[3], Michele Manfredda[3], Marco Zangrando [3,4], Nicola Mahne[3,4], Lorenzo Raimondi[3], Emanuele Pedersoli [3], Denys Naumenko[3] & Stefan Eisebitt[1,2]

Free-electron lasers (FELs) in the extreme ultraviolet (XUV) and X-ray regime opened up the possibility for experiments at high power densities, in particular allowing for fluence-dependent absorption and scattering experiments to reveal non-linear light–matter interactions at ever shorter wavelengths. Findings of such non-linear effects are met with tremendous interest, but prove difficult to understand and model due to the inherent shot-to-shot fluctuations in photon intensity and the often structured, non-Gaussian spatial intensity profile of a focused FEL beam. Presently, the focused beam is characterized and optimized separately from the actual experiment. Here, we present the simultaneous measurement of XUV diffraction signals from solid samples in tandem with the corresponding single-shot spatial fluence distribution on the actual sample. Our in situ characterization scheme enables direct monitoring of the sample illumination, providing a basis to optimize and quantitatively understand FEL experiments.

[1] Max-Born-Institut Berlin, Max-Born-Straße 2a, 12489 Berlin, Germany. [2] Institut für Optik und Atomare Physik, Technische Universität Berlin, Hardenbergstraße 36a, 10623 Berlin, Germany. [3] Elettra Sincrotrone Trieste S.C.p.A., Strada Statale 14, KM 163.5, 34012 Basovizza, TS, Italy. [4] Istituto Officina dei Materiali, Consiglio Nazionale delle Ricerche, 34149 Basovizza, TS, Italy. Correspondence and requests for materials should be addressed to M.S. (email: mschneid@mbi-berlin.de) or to S.E. (email: eisebitt@mbi-berlin.de)

The development of free-electron lasers for the extreme ultraviolet (XUV) and X-ray regime has been one of the major leaps in photon-based science in the last few decades. It enabled key advances in the study of ultrafast dynamics of excitations in matter with a unique combination of coherent femtosecond pulses, and ultrahigh fluences up to several J cm$^{-2}$ for XUV radiation, and several kJ cm$^{-2}$ in the hard X-ray regime[1,2]. In recent years, observations of non-linear effects in solids such as wave mixing[3,4], stimulated emission[5–7], and absorption saturation[8] have been reported. Conducting such experiments requires sophisticated control of the sample illumination. This includes the in situ control of the focus position and, possibly, the precise alignment of several free-electron laser (FEL) beams. After the experiment, an exact knowledge of the number of photons per unit time and area on the sample is crucial to interpret the measurements.

A number of well-established techniques exist to estimate these pivotal parameters. Gas monitor detectors are able to measure the total photon number in a single, few-femtosecond pulse[9], but cannot account for the intensity distribution within the focal spot on the sample. This distribution is typically measured separately from the actual experiment using wave-front sensing[10–12], ablative imprints[13,14], or by detecting the transmitted intensity through a small aperture or behind a sharp knife-edge scanned across the beam in the sample plane[2,15]. These approaches are highly invasive and cannot be performed in tandem with the majority of FEL experiments. They are in particular incompatible with all scattering experiments in the forward direction and cannot account for the finite acceptance of a sample smaller than the beam size or for the beam position on a larger and potentially inhomogeneous sample. This leads to significant uncertainties, especially in diffract-and-destroy experiments, where a new sample is aligned after every single shot[16–18].

In this work, we demonstrate the simultaneous measurement of the spatial fluence distribution on the sample in conjunction with the diffraction signal from a solid sample in a single-shot XUV FEL experiment. Our measurement scheme is derived from work on monolithically integrated gratings on carrier membranes[19] and from the theoretical treatment of zone-plate diffraction under off-axis illumination[20]. Via our integrated diffraction monitor design, we are able to map the incident photon distribution on the sample to the detector plane. There, the illumination is recorded simultaneously with the sample's scattering signal. This allows a precise control of the sample position during the experiment and yields reliable information on the sample illumination that is crucial for the interpretation of the data recorded.

## Results

### Role of the fluence estimate in non-linear experiments.
We demonstrate the importance of a precise fluence measurement for the interpretation of non-linear effects in Fig. 1. Here, we simulate a strongly focused, non-Gaussian FEL beam and consider three different estimates of the spatial fluence distribution $f$ in the sample plane $(\xi, \eta)$. The distributions represent, respectively, an accurate measurement (Fig. 1a), a blurred, low-resolution estimate as is typically the result of an aperture scan (Fig. 1b) and a constant estimate, where the shot energy is distributed uniformly over a certain area (Fig. 1c). The fluence histograms vary drastically for the different estimates, as shown in Fig. 1d. For each fluence distribution, we calculate the signal levels assuming a linear, power-law, or saturating fluence dependency (Fig. 1e–g, respectively). These non-linear relations occur for example in two-photon absorption or saturable absorption experiments. It is evident that, except for the linear case, an inaccurate fluence

assumption obfuscates some or all of the characteristic parameters of the effect under study. Thus, a correct interpretation of fluence-dependent measurements will only be possible with an accurate, in situ characterization of the incident photon distribution on the sample on a shot-by-shot basis.

**Grating design**. We consider an FEL diffraction experiment that detects scattered radiation as a function of momentum transfer **q** on a two-dimensional pixelated detector, as shown in Fig. 2. Note that this geometry also includes standard spectroscopy of the photon beam, where the beam at a selected momentum transfer (such as $q = 0$) is detected as a function of wavelength. In material and life sciences, thin membranes of $Si_3N_4$, Si, or polymers are commonly used to administer samples to the FEL beam. We equip these membranes with a grating structure, that gives rise to an additional scattering signal at a selected detector position[19]. The key idea of this work is to design the gratings such that each point on the sample surface diffracts the incoming light to a separate position on the detector while preserving the spatial relationship of the originating sample points. Figure 3 sketches the basic idea of our concept as a step-wise evolution from regular, to segmented, and finally to the spatially resolving gratings we discuss here.

We start with the following sinusoidal transmission function for a regular grating[21]:

$$t(\xi, \eta) = \frac{1}{2} + \frac{1}{2} \cos\left(\frac{2\pi}{p}\left(\xi\cos(\varphi) + \eta\sin(\varphi)\right)\right). \quad (1)$$

Here, $\xi$ and $\eta$ are spatial coordinates in the sample plane, while $p$ and $\varphi$ are the grating period and orientation angle, respectively. They are given by the position $(x, y, z_{det})$ of the grating's first diffraction order in the detector plane:

$$\varphi = \arctan(y/x), \quad (2)$$

$$p = \frac{\lambda}{\sin\left(\arctan\left(\sqrt{x^2 + y^2}/z_{det}\right)\right)}. \quad (3)$$

We turn $\varphi$ and $p$ into functions of the sample coordinates by setting

$$x = x_0 + m\xi, \quad (4)$$

$$y = y_0 + m\eta. \quad (5)$$

When inserted into Eq. (1), the result is a grating, the pitch and orientation of which varies continuously and that diffracts an image of its own illumination function, magnified by the dimensionless parameter $m$ and centered at $(x_0, y_0)$, to the detector. We note that the thereby obtained structures constitute segments of Fresnel zone plates (Supplementary Note 1).

Since Eqs (2) and (3) relate to far-field diffraction, the mapping is only valid if the detector is sufficiently far away from the sample to be in the Fraunhofer regime[21]. Specifically, this requires

$$z_{det} > 2\frac{w_0^2}{\lambda}, \quad (6)$$

where $w_0$ is the beam's waist size on the sample.

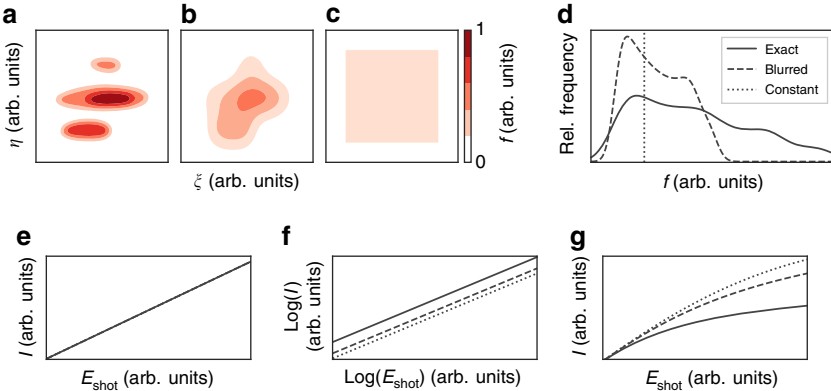

**Fig. 1** Examples of spatial fluence distribution estimates. Simulated two-dimensional maps of the spatial fluence distribution in the sample plane ($f(\xi, \eta)$, representing **a** an exact measurement, **b** a blurred, low-resolution measurement, and **c** a constant estimate. All maps sum to the same overall shot energy $E_{shot}$. **d** Fluence histograms of the three maps in **a**–**c**. **e**–**g** Simulated scattering signal $I(f)$ for increasing overall shot energy in the three fluence distributions under the assumption of, respectively, linear, quadratic, and saturating fluence dependency. Note the double logarithmic scale in **f**

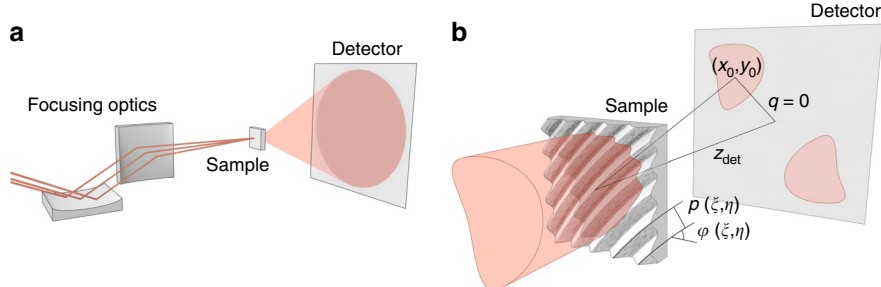

**Fig. 2** Experiment geometry and sample design. **a** An optical system focuses the incoming beam (red lines) onto the sample. Downstream, a 2D pixelated detector records the scattered radiation (red cone). **b** Enlarged view of the sample and the scattering geometry. The sample bears a suitably tailored, continuously varying grating with local periodicity $p(\xi, \eta)$ and local orientation angle $\varphi(\xi, \eta)$, where $\xi$ and $\eta$ are the coordinates in the sample plane. Incident light is diffracted away from the undeflected beam ($q = 0$) with a momentum transfer of $\pm\mathbf{q}(p, \varphi)$ according to the local grating parameters. We design the grating such that it maps an enlarged image of the incident illumination, centered around $\pm(x_0, y_0)$ in the detector plane at distance $z_{det}$.

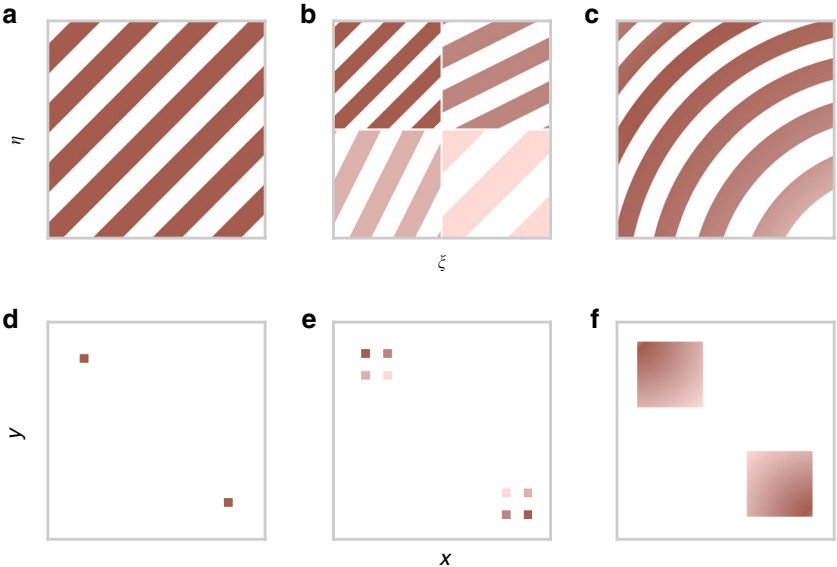

**Fig. 3** Schematic evolution of our grating design from regular gratings. The images show the real-space structures **a**–**c** and respective diffraction pattern **d**–**f**. **a** A regular grating diffracts incoming light to two symmetric points in Fourier-space **d** and hence reveals no spatial information on the illumination function. **b** A two-by-two segmented grating yields two symmetric sets of four diffraction spots **e**. The intensity of each spot is proportional to the illumination of the corresponding sample quadrant. This constitutes the most basic form of a spatially resolving beam profile monitor based on an integrated grating. **c** A grating with suitably varying period and orientation forms a magnified image of its own illumination in Fourier-space **f**. The colors in the real-space images indicate the local grating period and mark the corresponding points in the diffraction images

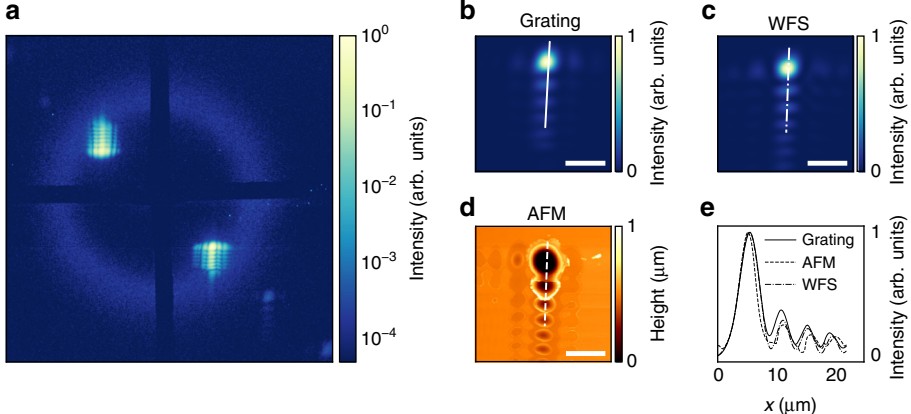

**Fig. 4** Single-shot spatial fluence distribution and primary sample signal. **a** Logarithmic false-color image of a single-shot diffraction pattern with sample signal (ring-shaped feature) and spatial fluence distributions. **b** Crop of the spatial fluence map from **a** on a linear intensity scale. **c** Spatial fluence distribution in the sample plane on a linear scale, extrapolated from a Hartmann wave-front sensor (WFS) measurement of a different FEL single-shot. **d** Atomic force micrograph (AFM) of a single-shot damage crater in the sample substrate. All scale bars correspond to 10 μm. **e** Line profiles along the indicated lines in **b**–**d**. To extract an intensity profile from **d**, we assume that the absorption in the substrate material follows a Beer–Lambert law. Neglecting thermal melting and redeposition, the damage crater topography then represents the surface of constant intensity at which the incident fluence is attenuated below the ablation threshold[13]

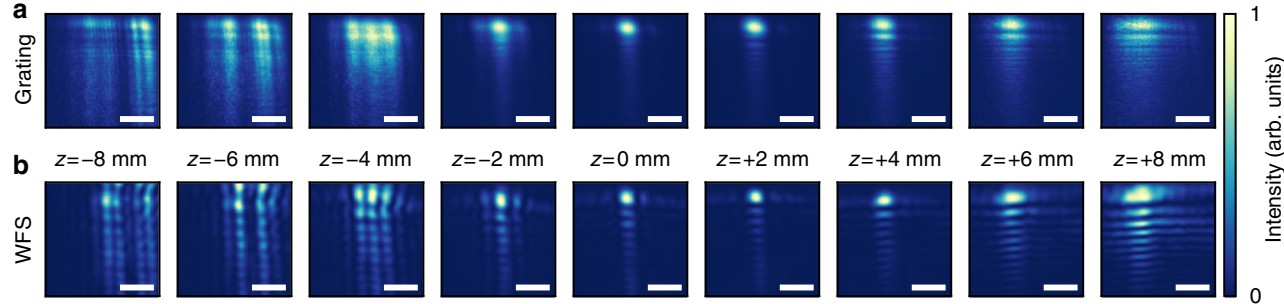

**Fig. 5** Spatial fluence distributions along the beam propagation axis. **a** Multi-shot image of the grating's positive diffraction order. The fluence distributions appear blurred due to spatial jitter. **b** Spatial fluence distribution calculated from a single-shot wave-front sensor measurement. The position z of the sample plane is given relative to the nominal focus position. Note that all images are individually normalized. The peak intensity drops rapidly when the sample moves out of the focus position. Scale bars correspond to 10 μm

In simulations with Gaussian beams, we observe that the following condition must simultaneously be satisfied:

$$\frac{z_{\text{det}}}{m} < 2\pi \frac{w_0^2}{\lambda}. \qquad (7)$$

This relation enforces that the illumination does not change drastically within a small number of grating periods, which would lead to errors in the diffracted fluence maps (see Supplementary Notes 1 and 2 for a detailed discussion).

**FEL experiment**. We present a single-shot XUV FEL diffraction pattern obtained in this fashion in Fig. 4a. The whole diffraction pattern consists of the ring-shaped primary sample signal (Methods) and the grating's positive and negative diffraction orders. Note that this particular experiment investigates the non-linear breakdown of the primary sample signal at high XUV fluences[22]. Under these special circumstances, the measured intensities of both signal contributions are not well matched.

The extracted map (Fig. 4b) reveals a complex focal spot with a bright central area and several side lobes of considerable intensity. Its brightest feature has a full width at half maximum (FWHM) of 3.9 μm and aggregates 58% of the shot energy. The recorded FEL single shot has a moderate pulse energy of 11 μJ. This corresponds to a peak fluence of 40 J cm$^{-2}$, which is well within the regime of previously reported non-linear XUV light–matter interactions[5,8,23]. Additionally, the beam's smallest features (1.9 μm FWHM) are clearly resolved by our grating monitor. The whole spatial distribution is in very good agreement with the independently obtained data for a different single shot via a Hartmann wave-front sensor measurement, shown in Fig. 4c. Both measurements are furthermore consistent with the atomic force microscopy image of a single-shot damage crater in the sample substrate, shown in Fig. 4d. The line scans in Fig. 4e further demonstrate that the relative intensities measured by our grating monitors agree well with the wave-front sensor and damage crater data. Deviations between the three independent measurements are due to uncertainties associated with each technique (e.g., melting and redeposition in the damage crater, choice of numerical propagation parameters for wave-front sensor image, wavelength, and distance-scaling of the diffraction image) and the fact that they originate from different FEL shots.

Figure 5 displays a series of spatial fluence distributions, recorded at various positions along the beam propagation axis. In order to record the diffraction signals (Fig. 5a) without risk of destroying the sample with a particularly high-powered single shot—as might occur due to random intensity fluctuations

inherent to the FEL source—we accumulate 2000 strongly attenuated shots per image. Thus, a perfect agreement with the wave-front sensor data (Fig. 5b) cannot be expected since the latter is extrapolated from a single-shot measurement. In particular, the accumulated images are blurred due to spatial jitter, that is, small, random changes of the beam pointing on a shot-to-shot basis. We are able to model the blurred, accumulated image at $z = 0$ (Fig. 5a, center) by applying a Gaussian filter to a destructive single-shot diffraction image (not shown), recorded at the same position. From this, we estimate the amount of spatial jitter in this particular case to be 3.5 μm (FWHM) in the horizontal, and 1.9 μm (FWHM) in the vertical direction. This is a significant fraction of the brightest feature's spatial extent of 3.9 μm. It is obvious, that—in addition to mapping the internal structure of the focus—the access to the spatial jitter of the beam on the sample on a single-shot basis is extremely valuable. This applies particularly to laterally inhomogeneous samples with spatially varying material composition, including particles sparsely dispersed on a membrane.

Throughout the series, grating monitor and wave-front sensor data generally agree well. As discussed, the principal cause for the differences to the single-shot wave-front measurement is spatial jitter of the FEL beam. Note that the fluence distribution changes substantially within a few millimeters along the beam axis. This is due to the finite size of the optical elements that act as limiting apertures and introduce diffraction artifacts into the beam. We remark that our in situ fluence-mapping approach can easily be used to track and optimize the sample position with respect to a focus both in the transverse direction as well as along the beam axis, e.g., when aligning samples for an experiment.

## Discussion

The attainable spatial resolution of our approach is closely linked to the manufacturing process. We use focused ion beam (FIB) milling to directly pattern the grating structures into the sample membrane (see Methods section for details on the manufacturing process). This fast and flexible method is capable of producing structure sizes down to a few tens of nanometers with low aspect ratios. As a rule of thumb, the necessary structure size (i.e., the grating half-pitch) to map a particular focal spot is about one-tenth of the spot size. Note that Eq. (7) enforces this condition (Supplementary Note 2 and Supplementary Fig. 1). Given the milling resolution of the FIB process, our approach is directly capable of mapping sub-micrometer focal spots in the XUV and soft X-ray regime.

Moving toward even shorter wavelengths and into the hard X-ray regime, the absorption contrast for most materials—and thereby the grating efficiency—diminishes. This necessitates higher aspect ratios for the grating structures and thus a more challenging manufacturing process. Furthermore, the achievable focal spots of the FEL beam are considerably smaller, extending into the sub-100 nm region[2,15]. Consequentially, smaller grating structures are necessary to satisfy Eq. (7).

Due to their close relationship to our grating monitors, it seems reasonable to consider recent progresses in hard X-ray zone-plate manufacturing for the discussion of these issues. Zone plates with 15 nm outer zone width have successfully been manufactured for hard X-ray radiation and are used in experiments with 9 keV photon energy[24]. With such manufacturing capabilities, it is feasible to directly transfer our concept to the hard X-ray regime and focal spot sizes on the order of 100 nm. Additionally, our concept does not require high diffraction efficiencies and aspect ratios can accordingly be smaller than for a zone plate. This makes it possible to utilize even smaller structures, and achieve sub-100 nm resolution. However, the complex manufacturing process might—at the current technological state—prohibit the time- and cost-efficient fabrication of a large number of samples for destructive studies. The actual limit will of course depend on the specific experiment, including photon energy, sample size and thickness, available detector space, and the experiment geometry.

The low absorption of hard X-ray radiation in most of the conceivable grating materials is greatly beneficial when transparent beam monitors are considered. In such cases, our grating concept could provide permanent and reliable in situ feedback of the beam position and spatial structure at critical beam-line positions, such as split-and-delay units[25,26] or intermediate focus stages[15]. This is further encouraged by the fact that spatial constraints in the experimental setup apply to a much lesser degree since high-vacuum conditions are usually not required. We note that no computational treatment of the measured fluence distributions is necessary. This makes the concept suitable for live monitoring, even at very high repetition rates.

Our fluence-mapping approach is a unique tool for true in situ, single-shot-capable monitoring of the fine structure of the sample illumination in transmission-type scattering experiments. It is, to our knowledge, the only approach that allows for a simultaneous, non-invasive mapping of the fluence distribution on the sample together with a scattering signal of interest. The approach provides an instantaneous online signal, which can be interpreted without any further computation, and can thus be used as instant feedback to align the upstream optical system. In the study of fluence-dependent phenomena, it provides crucial information for the correct interpretation of the data. The position and magnification of the photon-fluence map on the detector is, within the discussed constraints and the limits of the particular manufacturing process, freely selectable. Furthermore, the derivation of the grating formula can easily be adapted for diffraction experiments in reflection geometry. This makes our fluence-mapping approach compatible with a large variety of experiments and samples. Given these features, we expect this approach to become a valuable tool for alignment and optimization, and in particular for the study of non-linear light–matter interaction in the XUV and X-ray regime.

## Methods

**Diffraction experiment**. We perform small-angle scattering in transmission geometry at the FERMI@Elettra FEL source, using the DiProI end-station[27]. Focusing is provided via a bendable Kirkpatrik–Baetz optics[28]. A Princeton Instruments PI-MTE in-vacuum charge-coupled device (CCD) camera (2048 × 2048 pixel, 13.5 μm edge length) detects the scattered radiation 75–150 mm downstream of the sample. For the data in Fig. 4a, the sample–CCD distance is 75 mm. A cross-shaped beam-stop in front of the camera blocks the intense radiation in the forward direction and potential membrane edge scattering. We subtract dark images to flatten the image background. In the single-shot image, we manually remove a linear background that is due to a read-out artifact in the CCD wells that are read after the highest intensity occurs and de-noise the image by applying a Gaussian filter with 2 px width. The incident X-ray pulses are tuned to a wavelength of 20.8 nm (59.6 eV) with single-shot pulse energies ranging from 0.5 to 60 μJ. For accumulating measurements at a repetition rate of 10 Hz, solid-state filters and a gas absorber reduce the pulse energy to a range between 20 and 80 nJ. The Hartmann WFS—manufactured by Imagine Optique—is equipped with a 72 × 72 pinhole grid (pinhole diameter 60 μm, pitch 180 μm) and has a nominal accuracy of $\lambda/100$.

**Sample fabrication**. The samples consist of $Si_3N_4$ membranes of 30 nm thickness with 30–200 μm edge length. For the purpose of other experiments, a magnetic Co/Pt multi-layer is deposited on the membranes by DC magnetron sputtering. In the experiments reported here, this sample layer is in a labyrinth-like domain state with magnetization vectors parallel or anti-parallel to the FEL beam axis. At the selected photon wavelength, these domains give rise to a ring-shaped scattering signal on the CCD detector via the X-ray magnetic circular dichroism effect. We use a focused $Ga^+$ ion beam (FIB) to mill gratings directly into the $Si_3N_4$ membrane at 30 kV acceleration voltage and 93 pA beam current. For the milling process, the gratings are generated on a grid of 3500 × 3500 points with $x_0 = y_0 = 9.5$ mm, $\lambda = 20.8$ nm, $z = 150$ mm, and $m = 80$. In this particular case, milling the 35 × 35 μm grating with a nominal topographic amplitude of 2 nm takes 60 s.

**Data availability**. The data that support the findings of this study are available from the corresponding authors upon reasonable request.

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

## Acknowledgements

We thank the FERMI accelerator and laser teams for their great support during the experiments.

## Author contributions

M.S., C.M.G., B.P. and S.E. conceived the experiment. M.S. and C.M.G. prepared the samples. M.S., C.M.G., B.P., F.C., E.P. and D.N. conducted the scattering experiments. M.M., M.Z., N.M. and L.R. performed the wave-front sensor measurements. S.E. supervised the project. M.S. and S.E. wrote the manuscript with input from all the authors.

## Additional information

**Competing interests:** The authors declare no competing financial interests.

