## [Peer Review File · Nature Communications]

Reviewers' comments:

Reviewer #1 (Remarks to the Author):

The manuscript entitled "In-situ single-shot diffractive fluence mapping for X-ray free electron laser pulses", written by M. Schneider et al reported simultaneous measurement of a spatial profile of the focused FEL beam with FEL experiments.

Recently, intense FEL pulses in the XUV, soft x-ray, and hard x-ray regions have pioneered scientific frontiers with non-linear optical interactions. However, it is not straightforward to measure a spatial profile of the focused beam accurately, which produces considerable uncertainty in evaluation of the beam intensity.

In this manuscript, the authors have developed a method to enable shot-to-shot, non-invasive measurement of beam profile in parallel to conducting FEL experiments. They used a thin membrane with a grating structure, which gives an additional signal of the enlarged beam profile on the detector plane. By scanning a position of the membrane shot-to-shot, the beam profile can be recorded simultaneously with measurement of scattering signals from samples.

I think this method is unique, interesting, and potentially useful for experiments with intense XFEL pulses. However, I do not recommend to publish it to Nature Communication in the present form. One of the most critical points is that potentials and applicability of this method has not been clearly discussed. A particular interest to treat non-linear x-ray interactions with matters is to measure a profile of a tiny spot with a size of sub micron or less, rather than that of a several-micron beam presented in this study. Also, a capability to extend to shorter wavelength region, possibly down to hard x-rays, is critically important. Limitations and their reasons (for example, fabrication limit of gratings, relationship of grating thickness with focal depth, spectral broadening) should be discussed for clarifying potentials of this method.

As another issue, it seems not easy for a wide range of readers to understand the working principle of this method only with equations shown in this manuscript. I would encourage the authors to illustrate the basic principle, which will help intuitive understanding. Fig 2b is a too rough sketch for this purpose.

Finally, I would say that the beam parameters (i.e., the focal size of several microns, pulse energy of several ten joules at a photon energy of 60 eV) used in this test are "standard", and not highly attractive to researchers of non-linear x-ray optical science. Still, it could be evaluated as an important work if the authors can show appealing potentials.

Minor comments

- I do not see substantial importance of Fig. 1 and its explanations, which may be reduced or omitted.
- In addition to Ref. [4-6], two papers, Rohringer et al Nature 2012 and Yoneda et al Nature 2016, should be appeared in regarding with stimulated emission in the soft and hard x-ray regions.
- In addition to wavefront sensing [8-10] and ablation [11,12], a conventional knife-edge scan was applied for measuring the beam profile KB systems at SACLA, though invasive. Yumoto et al, Nature Photon 2013, and/or Mimura et al Nature Commun 2013 should be referred.

Reviewer #2 (Remarks to the Author):

The manuscript describes a novel method to greatly enhance the interpretation and control of measurements of non-linear light-matter interaction at XUV and X-ray wavelengths provided by the inherently fluctuating light pulses from free electron lasers. The method employs gratings akin to Fresnel-lenses fabricated onto the membranes used as sample holders in FEL diffraction experiments to provide a single-shot image of the laser focus at the sample on the CCD detector together with the diffraction signals from the sample proper. This is based on prior work of the authors in [16] where only integrated fluence was obtained in this manner, while this manuscript shows a substantial improvement as now the fluence distribution is accessible in single-shot. The paper demonstrates that the such obtained fluence distributions on the sample agree well with distributions obtained with other conventional methods which not allow an in-situ measurement, and also offers explanations and correction schemes for the observed discrepancies.

This paper is of great interest for one of the highlights of X-ray free electron laser applications, the study of non-linear interactions at X-ray wavelengths under extremely high fluence and will certainly help to improve the understanding in this area.

One point of criticism is that the title and more so the abstract of the paper suggest that a new method is demonstrated to simultaneously measure both diffraction signals from a sample of interest and also the fluence distribution at the sample by way of the imprinted grating on the sample substrate. Contrary to that claim, no actual diffraction signals from a real sample are shown, the only diffraction signals shown are the ones from the imprinted grating on the substrate. The data presented here demonstrates the feasibility of this in-situ method, and the location of the diffraction spots on the detector can be tailored to not interfere with an actual diffraction pattern from a real sample. Furthermore, in a previous publication [16] where only integrated single-shot fluence was measured, it was actually shown that both diffraction signals from a real sample and from the substrate grating can be distinguished. So the authors claim is certainly well funded, but it should at least be made clear what was actually demonstrated here, or additional data (most likely not immediately available) should be presented which shows in-situ single shot fluence maps and suitable diffraction patterns from a real sample.

The manuscript is very well written, structured and organized, and the submitted figures suitably support the motivation for this research, depict the experimental setup, and demonstrate the validity of the method. The materials in the methods section should enable reproduction of the method. The submitted supplemental material is also well suited to support the authors work and most of the claims.

Following are a list of further comments and suggestions to improve the paper.

- 1) Fig. 1 : Caption wording should be more clear to point out that the example is simulated data and not based on actual measurements (though this is clear from the main text).
- 2) Fig. 1g : Plot title missing parentheses in $1/1 + \exp(\dots)$
- 3) Fig. 2b : While Fig. 2a is referenced in the main text, 2b is not at all. Periodicity p is only mentioned in the figure caption, but appears in the main text unreferenced ($f=1/p$).
- 4) Page 5: The only quantified parameter in main text is the FEL position jitter which is not put into any context with other parameters (e.g. focus spot size, potential sample non-uniformity scale etc.) and by itself these numbers are not very informative or useful to the general reader. One has to look at the caption and the scale of Fig. 3 to gain some understanding of it.

Reviewer #3 (Remarks to the Author):

This manuscript reports a method to measure the 2D intensity distribution of X-ray free electron laser (XFEL) pulses. By using a soft XFEL with wavelength of 20.8 nm, it is demonstrated that the single- and multiple-shot intensity distributions can be mapped out. Although this work may be of interest to the XFEL community, a major limitation of this method is the requirement of a specially designed grating (Figure 2b). For soft XFELs, this works fine, but for hard XFELs, one will require a crystal with atomic scale lattices to match the short X-ray wavelengths, which will likely make the measurement more challenging.

Furthermore, it is claimed in the abstract that this represents “the first simultaneous measurement of diffraction signals from solid samples in tandem with the corresponding single-shot spatial fluence distribution on the actual sample”. However, the results presented in Figures 3 and 4 are preliminary. They show the single- and multiple-shot fluence distributions, but not the diffraction signals from a sample. In order to make a convincing case, I think it is necessary to show both the spatial fluence distribution and the diffraction signals from a meaningful sample.

Finally, it is stated in page 5 that “we find excellent agreement” between the measured and calculated fluence distributions in Figure 3. But after carefully comparing the images, I found they are different. I think the differences need to be explained.

In summary, I find the results reported in this manuscript are still preliminary. To make a convincing case, it is necessary to demonstrate the method with a more realistic sample. Furthermore, it is unclear to me whether this method will work for hard XFELs. Therefore, I cannot recommend its publication in Nature Communications.

Reviewer #4 (Remarks to the Author):

The presented paper is an interesting new way to use nanostructures and diffraction effects to imprint samples with a grating that could be used to characterize the focus on a shot-to-shot basis at FELs. The authors do a good job explaining why this characterization is necessary, and the method itself is innovative and interesting, and also opens up the door for new scientific measurements and analysis at FEL experiments. The method is not ideal, as it can suffer in case the scattering of the experimental sample is as large as the diffraction of the focal spot, and the authors list those few limitations in their work.

However, the paper, though interesting, should make a better case for itself when talking about experimental results. The figure 4 comparison with the wave-front method, for example, is not bad, but the authors themselves admit that it is not ideal on page 5, and that the Fraenhofer approximation for the analysis isn't fully satisfied. They also mention that this can be corrected if required. I would say that a publication in a journal as prestigious as Nature Communications requires the extra effort to be put in to either perform the experiment with the approximation satisfied, or to do the proper analysis to match the wavefront sensor data.

Lastly, it would be extremely useful to have a picture, graphic, or figure that shows what this diffracted focal spot looks like along with an actual experimental screen shot. Something where one can see both the diffracted focus spot and the data relevant to the experiment on the same screen. This would give us an idea of how easy to find and how easy to analyze this effect is. I, for one, have

some worry that the spot may be hard to find in a more complicated set of data, and it would be nice to see the authors explain by which procedure these spots are identified.

Reply to the reviewers' comments

We thank all four reviewers for their criticism and helpful feedback. In this document, we reproduce the individual comments (*italics*) and reply to them one by one.

A vertical line marks a direct reproduction of the respective changes in the revised manuscript.

As a preamble to our point-by-point reply, we'd like to make the following general remarks regarding the revision of our manuscript: We were able to prepare new samples and to record a new, practically aberration-free set of measurements at the FERMI FEL source. Furthermore, we now provide clear and simple relations that, if fulfilled, avert aberrations such as were present in the previous version of our manuscript. Both improvements enable us to focus much more on the discussion of our results, rather than of sources of error. We also included a detailed section on the scalability of our concept, both in terms of smaller spot-sizes and shorter wavelengths.

Reviewer 1

The manuscript entitled “In-situ single-shot diffractive fluence mapping for X-ray free electron laser pulses”, written by M. Schneider et al reported simultaneous measurement of a spatial profile of the focused FEL beam with FEL experiments. Recently, intense FEL pulses in the XUV, soft x-ray, and hard x-ray regions have pioneered scientific frontiers with non-linear optical interactions. However, it is not straightforward to measure a spatial profile of the focused beam accurately, which produces considerable uncertainty in evaluation of the beam intensity. In this manuscript, the authors have developed a method to enable shot-to-shot, non-invasive measurement of beam profile in parallel to conducting FEL experiments. They used a thin membrane with a grating structure, which gives an additional signal of the enlarged beam profile on the detector plane. By scanning a position of the membrane shot-to-shot, the beam profile can be recorded simultaneously with measurement of scattering signals from samples. I think this method is unique, interesting, and potentially useful for experiments with intense XFEL pulses. However, I do not recommend to publish it to Nature Communication in the present form.

1. *One of the most critical points is that potentials and applicability of this method has not been clearly discussed. A particular interest to treat non-linear x-ray interactions with matters*

is to measure a profile of a tiny spot with a size of sub micron or less, rather than that of a several-micron beam presented in this study.

The main text now discusses the achievable resolution on the basis of the minimum structure sizes given by our manufacturing process. From these limits we infer a resolution limit well below 1 μm given our particular manufacturing process. Alternative fabrication methods, e.g. based on electron-beam lithography, achieve better resolutions and could enable a resolution of 100 nm or better, as we point out in the discussion of hard x-rays.

The attainable spatial resolution of our approach is closely linked to the manufacturing process. We use focused ion beam (FIB) milling to directly pattern the grating structures into the sample membrane (see Methods section for details on the manufacturing process). This fast and flexible method is capable of producing structure sizes down to a few ten nanometers with low aspect-ratios. As a rule of thumb, the necessary structure size (i.e. the grating half-pitch) to map a particular focal spot is about one-tenth of the spot-size. Note that equation (7) enforces this condition (see Supplementary Notes). Given the milling resolution of the FIB process, our approach is directly capable of mapping sub-micrometer focal spots in the XUV and soft x-ray regime.

In support towards this claim, the smallest features of the focal spot in our new experimental data has a size of 1.9 μm (FWHM), as confirmed by wave-front sensor data and AFM scans of a damage crater. The data demonstrates that our grating monitor is easily capable to resolve all features of this particular spot (see updated Fig.3) and that smaller features are well within reach of the concept.

We note that several micrometer sized focal spots are relevant for non-linear light–matter interactions in the XUV regime, as is clearly supported by literature (e.g. refs [4, 7, 22] in the manuscript). In particular, these publications report spot sizes from 1.5 μm to 45 μm . The revised manuscript provides a better perspective on the field by directly linking the quantified spot parameters to these studies:

The extracted map (Fig. 3b) reveals a complex focal spot with a bright central area and several side lobes of considerable intensity. Its brightest feature has a full width at half maximum (FWHM) of 3.9 μm and aggregates 58 % of the shot energy. The recorded FEL single-shot has a moderate pulse energy of 11 μJ . This corresponds to a peak fluence of 40 J/cm^2 , which is well within the regime of previously reported non-linear XUV light–matter interactions [4, 7, 22].

2. Also, a capability to extend to shorter wavelength region, possibly down to hard x-rays, is critically important. Limitations and their reasons (for example, fabrication limit of gratings, relationship of grating thickness with focal depth, spectral broadening) should be discussed for clarifying potentials of this method.

We agree that this discussion would further broaden the interested audience. Therefore, our manuscript now includes a paragraph on the differences in the hard x-ray regime as compared to the experimentally demonstrated XUV case. This includes discussion of the relevant spot sizes and consequences of the diminishing absorption contrast. We reference recent literature on zone-plate manufacturing to illustrate the current technological limitations and their relationship to the performance of our concept.

When we consider our concept at shorter wavelengths, i.e. in the hard x-ray regime, three main distinctions to the demonstrated XUV case become apparent: i) The absorption contrast for most materials – and thereby the grating efficiency – diminishes. This necessitates higher aspect-ratios for the grating structures and thus a more challenging manufacturing process. ii) The achievable focal spots of the FEL beam are considerably smaller, extending into the sub-100 nm region [14, 15]. And iii) Spatial constraints in the experimental setup apply to a much lesser degree since high-vacuum conditions are usually not required and scattering angles for a given structure size are smaller. Thus, smaller grating structures are necessary to satisfy equation (7).

Due to their close relationship to our grating monitors, it seems reasonable to consider recent progresses in hard x-ray zone-plate manufacturing for the discussion of these issues. Zone-plates with 15 nm outer zone width have successfully been manufactured for hard x-ray radiation and are used, e.g. in experiments with 9 keV photon energy [23]. With such manufacturing capabilities, it is feasible to directly transfer our concept to the hard x-ray regime and focal spot sizes in the order of 100 nm. Additionally, our concept does not require high diffraction efficiencies and aspect ratios can accordingly be smaller than for a zone-plate. This makes it possible to utilize even smaller structures, and achieve sub-100 nm resolution. However, the complex manufacturing process might – at the current technological state – prohibit the time- and cost-efficient fabrication of a large number of samples for destructive studies. The actual limit will of course depend on the specific experiment, including photon energy, sample size and thickness, available detector space and the experiment geometry.

Nevertheless, we'd like to point out that also the XUV and soft x-ray communities alone are large. We feel that even if a certain method development would be restricted to that spectral range, this would not per se preclude publication in Nature Communications.

3. As another issue, it seems not easy for a wide range of readers to understand the working principle of this method only with equations shown in this manuscript. I would encourage the authors to illustrate the basic principle, which will help intuitive understanding. Fig 2b is a too rough sketch for this purpose.

The clear illustration of the basic working principle is of course a major objective of our manuscript. In response to the reviewer's recommendation, we added a schematic illustration (Fig. 2b) that shows a step-by-step evolution of our grating monitor concept. We believe that this will enable a more intuitive understanding of the basic idea.

4. Finally, I would say that the beam parameters (i.e., the focal size of several microns, pulse energy of several ten joules at a photon energy of 60 eV) used in this test are “standard”, and not highly attractive to researchers of non-linear x-ray optical science. Still, it could be evaluated as an important work if the authors can show appealing potentials.

The beam parameters in our study are typical for the FERMI FEL (e.g. ref. [26]) and comparable with the parameters reported in current literature on experiments pertaining to non-linear effects in this spectral range.

5. I do not see substantial importance of Fig. 1 and its explanations, which may be reduced or omitted.

Fig. 1 serves to illustrate problems that motivate this work. We acknowledge that the importance of these explanations is a very subjective matter and can be debated. In this particular case, we believe that our manuscript benefits from the figure and the corresponding explanations towards a clear motivation, as mentioned by reviewers #2 and #4. Thus, we would prefer to retain this section in the revised manuscript.

6. In addition to Ref. [4-6], two papers, Rohringer et al Nature 2012 and Yoneda et al Nature 2016, should be appeared in regarding with stimulated emission in the soft and hard x-ray regions.

We thank the reviewer for these suggestions. The publication of Yoneda et al. (Nature 2016) helps to provide a more complete view on recent experiments in non-linear x-ray – matter interactions and we now reference it accordingly. However, we believe that referencing Rohringer et al. (Nature 2012) could be somewhat misleading for the reader since the publication reports on experiments on Neon, i.e. with a gaseous sample. Our method aims at solid samples and we believe that keeping the distinction between these two classes of experiments is beneficial for a clear and concise manuscript.

7. In addition to wavefront sensing [8-10] and ablation [11,12], a conventional knife-edge scan was applied for measuring the beam profile KB systems at SACLA, though invasive. Yumoto et al, Nature Photon 2013, and/or Mimura et al Nature Commun 2013 should be referred.

We thank the reviewer for pointing out these relevant publications. The revised manuscript contains references to both papers at the suggested position as well as in the discussion of the hard x-ray situation.

This distribution is typically measured – separately from the actual experiment – using wavefront sensing [9–11], ablative imprints [12, 13], or by detecting the transmitted intensity

through a small aperture or behind a sharp “knife-edge” scanned across the beam in the sample plane [14, 15].

ii) The achievable focal spots of the FEL beam are considerably smaller, extending into the sub-100 nm region [14, 15].

Reviewer 2

The manuscript describes a novel method to greatly enhance the interpretation and control of measurements of non-linear light-matter interaction at XUV and X-ray wavelengths provided by the inherently fluctuating light pulses from free electron lasers. The method employs gratings akin to Fresnel-lenses fabricated onto the membranes used as sample holders in FEL diffraction experiments to provide a single-shot image of the laser focus at the sample on the CCD detector together with the diffraction signals from the sample proper. This is based on prior work of the authors in [16] where only integrated fluence was obtained in this manner, while this manuscript shows a substantial improvement as now the fluence distribution is accessible in single-shot. The paper demonstrates that the such obtained fluence distributions on the sample agree well with distributions obtained with other conventional methods which not allow an in-situ measurement, and also offers explanations and correction schemes for the observed discrepancies.

This paper is of great interest for one of the highlights of X-ray free electron laser applications, the study of non-linear interactions at X-ray wavelengths under extremely high fluence and will certainly help to improve the understanding in this area.

1. *One point of criticism is that the title and more so the abstract of the paper suggest that a new method is demonstrated to simultaneously measure both diffraction signals from a sample of interest and also the fluence distribution at the sample by way of the imprinted grating on the sample substrate. Contrary to that claim, no actual diffraction signals from a real sample are shown, the only diffraction signals shown are the ones from the imprinted grating on the substrate. The data presented here demonstrates the feasibility of this in-situ method, and the location of the diffraction spots on the detector can be tailored to not interfere with an actual diffraction pattern from a real sample. Furthermore, in a previous publication [16] where only integrated single-shot fluence was measured, it was actually shown that both diffraction signals from a real sample and from the substrate grating can be distinguished. So the authors claim is certainly well funded, but it should at least be made clear what was actually demonstrated here, or additional data (most likely not immediately available) should be presented which shows in-situ single shot fluence maps and suitable diffraction patterns from a real sample.*

This major point of criticism is now addressed with new experimental data that replaces the previously shown images (Figs. 3, 4).

The manuscript is very well written, structured and organized, and the submitted figures suitably support the motivation for this research, depict the experimental setup, and demonstrate the validity of the method. The materials in the methods section should enable reproduction of the method. The submitted supplemental material is also well suited to support the authors work and most of the claims.

Following are a list of further comments and suggestions to improve the paper.

2. Fig. 1 : Caption wording should be more clear to point out that the example is simulated data and not based on actual measurements (though this is clear from the main text).

We agree that the previous caption wording was unclear in this point and changed the caption text accordingly.

Figure 1: Examples of spatial fluence distribution estimates. Simulated two-dimensional spatial fluence maps in the sample plane (ξ, η) . [...]

3. Fig. 1g : Plot title missing parentheses in $1/1 + \exp(\dots)$

We thank the reviewer for spotting this typo. Parentheses are now added to correct the formula.

4. Fig. 2b : While Fig. 2a is referenced in the main text, 2b is not at all.

Former Fig. 2b (now Fig. 2c) shows the geometry and notations for the derivation of the grating formula and is now referenced accordingly in the main text.

The scattering geometry, including the notations we use in the following derivation of the grating formula are shown in Fig. 2c.

5. Periodicity p is only mentioned in the figure caption, but appears in the main text unreferenced ($f=1/p$).

We changed the derivation of the grating formula to not make use of the grating frequency f in favor of the consistent use of the grating period p . The revised manuscript references all used parameters.

6. *The only quantified parameter in main text is the FEL position jitter which is not put into any context with other parameters (e.g. focus spot size, potential sample non-uniformity scale etc.) and by itself these numbers are not very informative or useful to the general reader. One has to look at the caption and the scale of Fig. 3 to gain some understanding of it.*

In the revised manuscript, we quantify the spot-size, the fraction of shot energy in the main feature, the size of the minor features and the spatial jitter. We calculate the peak fluence and provide references that relate these parameters to experimental demonstrations of non-linear light–matter interactions in recent literature. We now discuss the amount of spatial jitter in relation to the size of the dominant spot features.

The extracted map (Fig. 3b) reveals a complex focal spot with a bright central area and several side lobes of considerable intensity. Its brightest feature has a full width at half maximum (FWHM) of $3.9\ \mu\text{m}$ and aggregates 58 % of the shot energy. The recorded FEL single-shot has a moderate pulse energy of $11\ \mu\text{J}$. This corresponds to a peak fluence of $40\ \text{J}/\text{cm}^2$, which is well within the regime of previously reported non-linear XUV light–matter interactions [4,7,22]. Additionally, the beam’s smallest features ($1.9\ \mu\text{m}$ FWHM) are clearly resolved by our grating monitor.

We are able to model the blurred, accumulated image by applying a Gaussian filter to the single-shot image (not shown). From this, we estimate the amount of spatial jitter in this particular case to be $3.5\ \mu\text{m}$ (FWHM) in the vertical, and $1.9\ \mu\text{m}$ (FWHM) in the horizontal direction. This is a significant fraction of the brightest feature’s spatial extent of $3.9\ \mu\text{m}$.

Reviewer 3

This manuscript reports a method to measure the 2D intensity distribution of X-ray free electron laser (XFEL) pulses. By using a soft XFEL with wavelength of 20.8 nm, it is demonstrated that the single- and multiple-shot intensity distributions can be mapped out.

1. *Although this work may be of interest to the XFEL community, a major limitation of this method is the requirement of a specially designed grating (Figure 2b). For soft XFELs, this works fine, but for hard XFELs, one will require a crystal with atomic scale lattices to match the short X-ray wavelengths, which will likely make the measurement more challenging.*

Provided that the experimental geometry allows to increase the detector distance (z_{det}), the required grating period to spatially resolve a beam footprint is determined solely by the spatial extent of that particular photon beam on the sample. Even for the smallest reported focal spots, i.e. in the sub-100 nm regime, this necessitates grating structures on the order of 10 nm. Such

structure sizes are within reach of current manufacturing capabilities (see also our reply to reviewer #1's second comment). We acknowledge that for diffraction signals from atomic-scale lattices, the diffraction signal of such an artificial grating might not fit on the same detector and that a separate detector at greater distance might be necessary. We believe that the revised manuscript addresses this more clearly in the discussion of the derived grating formula:

This relation enforces that the illumination does not change drastically within a small number of grating periods, which would lead to errors in the diffracted fluence maps (see section "Simulations: beam footprint" in the Supplementary Notes for a detailed discussion).

And further in the paragraph on the resolution limit for both, soft and hard x-ray beams:

As a rule of thumb, the necessary structure size (i.e. the grating half-pitch) to map a particular focal spot is about one-tenth of the spot-size.

As well as in the Supplementary Notes. As pointed out in our response to the comments of reviewer #2, we have now included a discussion on the scalability of our approach to harder x-rays.

2. Furthermore, it is claimed in the abstract that this represents "the first simultaneous measurement of diffraction signals from solid samples in tandem with the corresponding single-shot spatial fluence distribution on the actual sample". However, the results presented in Figures 3 and 4 are preliminary. They show the single- and multiple-shot fluence distributions, but not the diffraction signals from a sample. In order to make a convincing case, I think it is necessary to show both the spatial fluence distribution and the diffraction signals from a meaningful sample.

This is the same issue raised by Reviewer #2 in point 1. As discussed above, we have recorded additional data to this effect, which is presented in the revised version of the manuscript.

3. Finally, it is stated in page 5 that "we find excellent agreement" between the measured and calculated fluence distributions in Figure 3. But after carefully comparing the images, I found they are different. I think the differences need to be explained.

We recorded our new experimental data in a geometry without these aberrations.

4. In summary, I find the results reported in this manuscript are still preliminary. To make a convincing case, it is necessary to demonstrate the method with a more realistic sample.

This concern is likewise addressed by the new measurements. We now demonstrate our concept using a realistic diffraction sample that is additionally equipped with a grating monitor as

described in the manuscript.

5. *Furthermore, it is unclear to me whether this method will work for hard XFELs.*

We discuss the applicability and possible limitations of our concept in the hard x-ray regime in the revised manuscript. We refer to our reply to the same concern raised by reviewer #1 in point 2.

Therefore, I cannot recommend its publication in Nature Communications.

Reviewer 4

The presented paper is an interesting new way to use nanostructures and diffraction effects to imprint samples with a grating that could be used to characterize the focus on a shot-to-shot basis at FELs. The authors do a good job explaining why this characterization is necessary, and the method itself is innovative and interesting, and also opens up the door for new scientific measurements and analysis at FEL experiments. The method is not ideal, as it can suffer in case the scattering of the experimental sample is as large as the diffraction of the focal spot, and the authors list those few limitations in their work.

1. *However, the paper, though interesting, should make a better case for itself when talking about experimental results. The figure 4 comparison with the wave-front method, for example, is not bad, but the authors themselves admit that it is not ideal on page 5, and that the Fraunhofer approximation for the analysis isn't fully satisfied. They also mention that this can be corrected if required. I would say that a publication in a journal as prestigious as Nature Communications requires the extra effort to be put in to either perform the experiment with the approximation satisfied, or to do the proper analysis to match the wavefront sensor data.*

The new experimental data that we show in the revised manuscript is recorded with the small-angle approximation satisfied. We now explicitly state the necessary conditions for a proper experiment in the revised manuscript.

Since equations (2) and (3) relate to far-field diffraction, the mapping is only valid if the detector is sufficiently far away from the sample to be in the Fraunhofer regime [21], i.e.

$$z_{\text{det}} > 2 \frac{w_0^2}{\lambda},$$

where w_0 is the beam's waist size on the sample.

In simulations with Gaussian beams we observe, that the following condition must simul-

taneously be satisfied:

$$\frac{z_{\text{det}}}{m} < 2\pi \frac{w_0^2}{\lambda}.$$

This relation enforces that the illumination does not change drastically within a small number of grating periods, which would lead to errors in the diffracted fluence maps (see section “Simulations: beam footprint” in the Supplementary Notes for a detailed discussion).

This allows us to omit describing the pertaining aberrations in favor of a more detailed discussion of the experimental results and the scalability to smaller focal sizes and harder x-rays, which we believe is of greater interest to the readership.

2. Lastly, it would be extremely useful to have a picture, graphic, or figure that shows what this diffracted focal spot looks like along with an actual experimental screen shot. Something where one can see both the diffracted focus spot and the data relevant to the experiment on the same screen. This would give us an idea of how easy to find and how easy to analyze this effect is. I, for one, have some worry that the spot may be hard to find in a more complicated set of data, and it would be nice to see the authors explain by which procedure these spots are identified.

Fig. 3a now shows a sample signal and the diffracted fluence map of the illuminating beam on the sample. The caustic scan through the focal position in Fig. 4 likewise shows actual experimental data. In fact, such scans of the grating along the beam – with a live image of the illumination profile as described in our manuscript – have, on the basis of our work, become a standard method to characterize the sample alignment relative to the focus at the DiProI end-station of FERMI.

REVIEWERS' COMMENTS:

Reviewer #1 (Remarks to the Author):

I found the revised version was much improved with reflecting comments from me and other reviewers. Thus I basically would like to support publication of the article in Nature Communications, while I attach one additional comment below:

p.2: "..., ultrahigh fluence up to several J/cm^2 , ..."

A typical fluence with focused beam with hard X-ray FEL reaches $100 \text{ uJ}/1 \text{ um}^2$, which corresponds to $10 \text{ kJ}/\text{cm}^2$. Thus it would be inappropriate to limit up to "several J/cm^2 ".

Reviewer #2 (Remarks to the Author):

Report on NCOMMS-17-09922A, In-situ single-shot diffractive fluence mapping for X-ray free-electron laser pulses

With their revised manuscript the authors have satisfactorily addressed the concerns raised previously by the reviewers and they present a significantly improved paper acceptable for publication in Nature Communications.

One remark left is that in the Methods section on page 11 the clause "solid-state filters and a gas absorber reduce the pulse energy to 20 nJ to 80 nJ" is not immediately understandable. Maybe say "... pulse energy to a range between 20 nJ and 80 nJ".

Reviewer #3 (Remarks to the Author):

The authors have presented new experimental results and fully addressed my points. I think the revised manuscript is suitable for Nature Communications.

Reviewer #4 (Remarks to the Author):

The authors have done an excellent job addressing the comments from the various reviewers, and have altered the original text appropriately to address our concerns. Therefore, I recommend that Nature Communications publishes the article as it is.

Reply to the reviewers' comments

We thank all four reviewers for their feedback and their favorable reviews. In this document, we reproduce the individual comments (*italics*) and reply to them one by one.

A vertical line marks a direct reproduction of the respective changes in the revised manuscript.

Reviewer 1

1. *I found the revised version was much improved with reflecting comments from me and other reviewers. Thus I basically would like to support publication of the article in Nature Communications, while I attach one additional comment below: p.2: "ultrahigh fluence up to several J/cm², ..." A typical fluence with focused beam with hard X-ray FEL reaches 100 $\mu\text{J}/1 \mu\text{m}^2$, which corresponds to 10 kJ/cm². Thus it would be inappropriate to limit "up to several J/cm²".*

We agree that a distinction between the hard- and soft x-ray/ XUV regime is necessary here. The revised sentence now reads:

It enabled key advances in the study of ultrafast dynamics of excitations in matter with a unique combination of coherent femtosecond pulses, and ultrahigh fluences up to several J cm^{-2} for XUV radiation, and several kJ cm^{-2} in the hard x-ray regime.

Reviewer 2

1. *With their revised manuscript the authors have satisfactorily addressed the concerns raised previously by the reviewers and they present a significantly improved paper acceptable for publication in Nature Communications.*

One remark left is that in the Methods section on page 11 the clause "solid-state filters and a gas absorber reduce the pulse energy to 20 nJ to 80 nJ" is not immediately understandable. Maybe say "... pulse energy to a range between 20 nJ and 80 nJ".

We agree that the wording was unnecessarily hard to understand and changed the sentence as suggested by the reviewer.

Reviewer 3

1. The authors have presented new experimental results and fully addressed my points. I think the revised manuscript is suitable for Nature Communications.

Reviewer 4

1. The authors have done an excellent job addressing the comments from the various reviewers, and have altered the original text appropriately to address our concerns. Therefore, I recommend that Nature Communications publishes the article as it is.